# Novel Screen-Printed Sensor with Chemically Deposited Boron-Doped Diamond Electrode: Preparation, Characterization, and Application

**DOI:** 10.3390/bios12040241

**Published:** 2022-04-13

**Authors:** Oleksandr Matvieiev, Renáta Šelešovská, Marian Vojs, Marián Marton, Pavol Michniak, Vojtěch Hrdlička, Michal Hatala, Lenka Janíková, Jaromíra Chýlková, Jana Skopalová, Petr Cankař, Tomáš Navrátil

**Affiliations:** 1Institute of Environmental and Chemical Engineering, Faculty of Chemical Technology, University of Pardubice, Studentská 573, 532 10 Pardubice, Czech Republic; oleksandr.matvieiev@student.upce.cz (O.M.); lenka.janikova@upce.cz (L.J.); jaromira.chylkova@upce.cz (J.C.); 2Institute of Electronics and Photonics, Faculty of Electrical Engineering and Information Technology, Slovak University of Technology in Bratislava, Ilkovičova 3, 812 19 Bratislava, Slovakia; marian.vojs@stuba.sk (M.V.); marian.marton@stuba.sk (M.M.); pavol.michniak@stuba.sk (P.M.); 3J. Heyrovsky Institute of Physical Chemistry of the Academy of Sciences of the Czech Republic, Dolejškova 3, 182 23 Prague, Czech Republic; vojtech.hrdlicka@jh-inst.cas.cz; 4Department of Graphic Arts Technology and Applied Photochemistry, Faculty of Chemical and Food Technology, Slovak University of Technology in Bratislava, Radlinského 9, 812 37 Bratislava, Slovakia; michal.hatala@stuba.sk; 5Department of Analytical Chemistry, Faculty of Science, Palacký University in Olomouc, 17. listopadu 1192/12, 779 00 Olomouc, Czech Republic; jana.skopalova@upol.cz; 6Department of Organic Chemistry, Faculty of Science, Palacký University in Olomouc, 17. listopadu 1192/12, 779 00 Olomouc, Czech Republic; petr.cankar@upol.cz

**Keywords:** screen-printed sensor, boron-doped diamond electrode, preparation, characterization, electrochemical properties, analytical application, lornoxicam

## Abstract

New screen-printed sensor with a boron-doped diamond working electrode (SP/BDDE) was fabricated using a large-area linear antenna microwave chemical deposition vapor system (LA-MWCVD) with a novel precursor composition. It combines the advantages of disposable printed sensors, such as tailored design, low cost, and easy mass production, with excellent electrochemical properties of BDDE, including a wide available potential window, low background currents, chemical resistance, and resistance to passivation. The newly prepared SP/BDDEs were characterized by scanning electron microscopy (SEM) and Raman spectroscopy. Their electrochemical properties were investigated by cyclic voltammetry and electrochemical impedance spectroscopy using inner sphere ([Fe(CN)_6_]^4−/3−^) and outer sphere ([Ru(NH_3_)_6_]^2+/3+^) redox probes. Moreover, the applicability of these new sensors was verified by analysis of the anti-inflammatory drug lornoxicam in model and pharmaceutical samples. Using optimized differential pulse voltammetry in Britton–Robinson buffer of pH 3, detection limits for lornoxicam were 9 × 10^−8^ mol L^−1^. The oxidation mechanism of lornoxicam was investigated using bulk electrolysis and online electrochemical cell with mass spectrometry; nine distinct reaction steps and corresponding products and intermediates were identified.

## 1. Introduction

The development of new sensors/electrodes/systems for analysis of biologically active compounds (BACs) and their metabolites, important from the point of views of medical, agricultural, environmental, or food safety, is one of the most important trends in modern analytical chemistry. At present, highly sophisticated, expensive, time-consuming, and labor-intensive spectrometric and separation methods are usually used for these purposes. However, their high investment and running costs complicate their use for large-scale monitoring and screening purposes and eventually for field applications, on-site monitoring, point-of-care devices, etc. For many electrochemically active substances, these methods can be successfully substituted by electrochemical methods whose main advantages can be characterized by low investment and running costs, simplicity, portability, and easy miniaturization. Their application in complex environmental or biological samples is sometimes limited by sensitivity and selectivity. Therefore, great attention has been focused on new electrode materials as well as on surface pretreatment or modification of traditional bare electrodes for improving their sensing properties. Another important direction is the development of novel screen-printed sensors (SPEs) which enable rapid and accurate in situ analyses and the development of portable devices for quantitation of pollutants, drugs, pesticides, biomolecules, antigens, microorganisms, enzymes, and many other BACs [1,2,3,4,5,6,7,8]. Their advantages include the possibility of tailoring their composition, modification, variable shape for a particular purpose, and the possibility of easy mass production, allowing the use as disposable sensors [9,10].

The main factor determining and limiting the electrochemical properties and applicability of the SPE is the material of the working electrode. In addition to conventional carbon and metal electrodes, papers describing the application of printed sensors with a working boron-doped diamond electrode (BDDE) have recently begun to be published [11]. The BDDE was introduced into electroanalytical practice almost 30 years ago [12,13,14] and established itself very well due to its excellent electrochemical properties, namely, a wide available potential window (in some cases more than 3 V), low background current, good chemical resistance, low current noise, resistance to passivation, and high hardness [15,16,17,18]. Diamond films are commonly prepared by chemical vapor deposition (CVD), which uses hot filaments or microwave plasma sources [19]. BDDEs have been used in the analysis of many substances important for the protection of the environment and human health. Voltammetric methods have been developed for the determination of a variety of environmental pollutants, health-hazardous substances, drugs, tumor markers, and others. The applicability of BDDE is described in several review articles [20,21,22,23,24]. Moreover, the modification of the BDDE surface can lead to further improvement of its properties and application possibilities [25,26,27]. Combining the technology of printed sensors with their specific advantages and a BDDE with its unique electrochemical properties can lead to a significant improvement or expansion of the application possibilities.

In general, the term SPE refers to sensors whose essence is the use of screen printing in their production. They consist of a chemically inert non-conductive substrate (ceramic, plastic (polyethylene terephthalate, polyamide, polyester), aluminum, glass, alumina composite) on whose surface three or more electrodes are pressed using functional liquid compositions in the form of dispersions or solutions [28]. The main advantage of SPEs, compared to the sensors made by conventional CVD production processes, is the additive printing process. This can be characterized as the application of individual layers of a precisely defined shape successively to each other using successive printing units. In the case of SPE, it is also possible to use rigid or flexible substrates for the preparation of disposable sensors. Other undeniable advantages in addition to a more flexible, faster, and more economical production process are the simplicity of adjusting the composition, thickness, and area of the electrode, statistical validation of experimental results thanks to replicated electrodes, and the possibility of easy modification of the electrode depending on the analyte. This modification can be realized in two ways, namely, by changing the composition of the compositions directly during their production by incorporating different substances or by additional surface treatment by applying different films. The most common material used to make a working electrode is carbon in its various forms and modifications [29,30], such as graphite and carbon black [31], graphene [32], graphene oxide [33], carbon nanotubes [34], and others. Different materials used alone or as carbon electrode modifiers in the case of non-enzymatic sensors include, in particular, metals [7,35,36], their oxides, or combinations thereof [37,38].

In this work, novel screen-printed sensors with boron-doped diamond electrodes were produced by combining two techniques, a large-area linear antenna microwave chemical vapor deposition system and a screen-printing technique. By combining these techniques, we are able to prepare a simple and cheap three-electrode system integrating highly-stable BDD electrodes (working and counter electrode) and Ag/AgCl quasi reference electrode. This system fulfills the concept of low-cost sensors for long-term on-line monitoring of a wide range of chemicals in harsh environmental analysis in static or flow systems. In addition, it can successfully serve as a disposable sensor in the analysis of biological samples (e.g., body fluids) in the so-called point-of-care analysis.

## 2. Materials and Methods

### 2.1. Chemicals

The stock solution of 0.1 mol L^−1^ KCl was prepared by dissolution of the appropriate amount of powder (Ing. Petr Švec-PENTA s.r.o., Prague, Czech Republic) in the distilled water and the standard solutions of 6.25 × 10^−4^ and 2.5 × 10^−3^ mol L^−1^ K_3_[Fe(CN)_6_] and [Ru(NH_3_)_6_]Cl_3_ (both of purity ≤99%, Sigma-Aldrich, Merck, Prague, Czech Republic) were prepared in the solution of 0.1 mol L^−1^ KCl. Britton–Robinson buffer (BRB) consisting of the mixture of an acidic (H_3_PO_4_, H_3_BO_3_, and CH_3_COOH (0.4 mol L^−1^), Penta-Švec, Czech Republic) and alkaline component (NaOH (0.2 mol L^−1^), Penta-Švec, Czech Republic) was used as a supporting electrolyte. The stock solution of 0.001 mol L^−1^ lornoxicam (≥98%, Sigma-Aldrich) was prepared by dissolution of its powder in methanol (Penta-Švec, Czech Republic) and stored in a refrigerator (+4 °C) without light access. Solutions of lower concentrations were prepared daily fresh by dilution with a supporting electrolyte.

### 2.2. Instrumentation

Voltammetric measurements were performed using Autolab PGSTAT204 (Metrohm Autolab, Utrecht, Netherlands) equipped with Nova 2.1 software. In the case of the conventional three-electrode arrangement of the electrochemical cell, bulk BDDE (BioLogic, Seyssinet-Pariset, France, surface area of 7.07 mm^2^, inner diameter of 3 mm, resistivity of 0.075 Ω cm with B/C ratio during deposition 1000 ppm) was applied as a working electrode (WE), saturated silver/silver chloride electrode (Ag|AgCl|KCl(sat.)) as a reference (RE), and platinum wire as a counter electrode (CE, both Monokrystaly, Turnov, Czech Republic). Additionally, the following two types of printed sensors have been used: (i) commercially available SPE (SP/BDDE, Metrohm Autolab (DropSens), Utrecht, Netherlands) with BDD WE (surface area 10.17 mm^2^, inner diameter 3.6 mm, B/C was not specified), carbon CE, and silver RE and (ii) laboratory-prepared SPE (LM-SP/BDDE) consisted of BDD as WE as well as CE and Ag|AgCl RE, when 3LM-sensor types with different working electrode surfaces were tested (WE surface area 0.785, 3.14, and 7.07 mm^2^, inner diameter of 1, 2, and 3 mm, B/C 312 500 in the gas phase, the resistivity of 0.017 Ω cm). All tested sensors are shown in Appendix A.

The Autolab PGSTAT128 N potentiostat (Metrohm Autolab, Utrecht, Netherlands) was employed for controlled potential electrolysis with a carbon fiber brush electrode (CFBE) [39]. The three-electrode system was completed with a reference saturated calomel electrode (SCE) and platinum auxiliary electrode, which was placed in a cathodic compartment separated by a glass frit from the anodic space containing the sample solution.

An Acquity UPLC system (Waters, Milford, MA, USA) with PDA detector and mass spectrometric detector (QDA) equipped with heated electrospray ionization (HESI) and quadrupole analyzer were used for the analysis of electrolyzed solutions. An ADLC1 potentiostat (Laboratorní přístroje, Prague, Czech Republic) with Model 5021A conditioning cell (ESA, Chelmsford, MA, USA) containing porous graphite working electrode, Pd counter and a Pd/H_2_ reference electrode, NE-1002X syringe pump (New Era Pump Systems, Farmingdale, NY, USA), and Agilent 1100 Series LC/MSD Trap (Agilent Technologies, Palo Alto, CA, USA) with electrospray ionization (ESI) in the positive and negative mode were employed for on-line EC-MS experiments.

Electron microscopy (JEOL 7500f, Tokio, JP, 45° angle view) was used to investigate the surface morphology and thickness of the BDD film from cross-section views. The chemical structure of the deposited films was evaluated by Raman spectroscopy (633 nm Dilor system, 5 μm spot diameter, and 325 nm Spectroscopy & Imaging, Warstein, Germany, 2 μm spot diameter). Hall measurements (4-point van der Pauw method) were applied for the determination of electrical properties (resistivity and carrier concentration), and samples were contacted by vacuum evaporation of 10 nm Cr and 100 nm Au multilayer pads. Measurements were carried out at room temperature.

The Accumet AB150 pH-meter (Fisher Scientific, Pardubice, Czech Republic) and the Bandelin Sonorex ultrasonic bath (Schalltec GmbH, Allmendingen, Germany) were used for various solutions preparation. Deionized water (conductivity < 0.05 μS cm^−1^) was prepared in a Millipore Mili plus Q system, USA.

Parameters of calibration curves and confidence intervals were calculated at the significance level of 0.05 using QCExpert software (TriloByte, Staré Hradiště, Czech Republic), MS Excel (Microsoft CZ, Prague, Czech Republic), and OriginPro 9.0. (OriginLab Corporation, Northampton, US). The limit of detection (LOD) was calculated from the calibration dependences as three times the standard deviation of an intercept divided by the slope [40,41].

### 2.3. Procedures

#### 2.3.1. Preparation of Screen-Printed Sensors with Chemically Deposited BDDE

All the individual steps of the preparation of the screen-printed sensor with chemically deposited boron-doped diamond electrode (LM-SP/BDDE) are illustrated in Figure 1. Depositions of boron-doped diamond were carried out in the linear antenna MWCVD reactor (Cube 300, Scia Ltd., Chemnitz, Germany) using 6 kW of microwave power for 30 h at 590 °C substrate temperature and 30 Pa pressure. The concentration of trimethyl borate (TMBT, Sigma-Aldrich, ≥99.0%) evaporated and introduced into the chamber was 1%, and CO_2_ concentration was 0.2% with respect to the background hydrogen. The resulting B/C ratio in the TMBT/CO_2_/H_2_ gas mixture was 312,500 ppm.

The silver electrode and the insulating layer were printed by a screen-printing technique. For the design definition, a stencil prepared in a photochemical way using a positive film template and a light-sensitive emulsion FOTECOAT 1019 BLUE (SPT Sales + Marketing, Heidelberg, Greece) was used. The printing process was realized using a semi-automatic printing machine TY-600H (ATMA, Taoyuan City, Taiwan) equipped with a vacuum table for substrates fixation. A polyurethane squeegee SERILOR HR1 P0 85 °Sh (Fimor, Le Mans, France) was used.

For the reference electrode (RE) preparation, silver particles containing AST6025 printing paste (SunChemical, Parsippany, NJ, USA) and polyester mesh with a mesh count of 71 threads cm^−1^ (SEFAR, Heiden, Switzerland) were used. Two layers were printed by the wet-on-wet method. To support the leveling process, the layer was left after printing at room temperature for 5 min and subsequently dried in a UN 55 laboratory oven (MEMMERT, Schwabach, Germany) at 150 °C for 30 min. The transformation of Ag to Ag|AgCl was carried out by chlorination process—chronoamperometry with RE connected as WE in constantly stirred 0.1 mol L^−1^ KCl solution. The applied voltage was +700 mV for 30 s.

A silicone-based screen-printing paste with mineral filler 240-SB (FERRO, Mayfield Heights, OH, USA) was used for the insulation layer preparation. The printing of two layers by the wet-on-wet method was carried out using a polyester mesh with a mesh count of 32 threads cm^−1^ (SEFAR, Heiden, Switzerland). After a leveling support (5 min at room temperature), the layer was dried in a UN 55 laboratory oven (MEMMERT, Schwabach, Germany) at 150 °C for 120 min.

#### 2.3.2. Electrochemical Characterization of Tested Sensors

Before starting work, bulk BDDE was always activated by performing 20 cyclic voltammograms in the potential range from initial potential (*E*_in_) of −1500 to switching potential (*E*_switch_) of +2200 mV at scan rate (*v*) of 100 mV s^−1^ directly in the supporting electrolyte used. It was found that it was not necessary to reactivate the electrode surface or regenerate it in any way between individual experiments. The printed sensors were also activated before use, but only enough CV cycles were applied to stabilize the background (10 cycles maximum).

Cyclic voltammetry (CV) was used for electrochemical characterization of tested sensors using conventional redox markers. The applied parameters for [Fe(CN)_6_]^4−/3−^ were as follows: *E*_in_ = +1000 mV (BDDE), +600 mV (SP/BDDE), and +600 mV (LM-SP/BDDE), *E*_switch_ = −600 mV (BDDE), −550 mV (SP/BDDE), and −350 mV (LM-SP/BDDE), *v* = 100 mV s^−1^. The applied parameters for [Ru(NH_3_)_6_]^2+/3+^ were as follows: *E*_in_ = +300 mV (BDDE) and +100 mV (SP/BDDE, LM-SP/BDDE), *E*_switch_ = −400 mV (BDDE) and −550 mV (SP/BDDE, LM-SP/BDDE), *v* = 100 mV s^−1^.

Electrochemical impedance spectroscopy (EIS) experiments were performed in the frequency range from 10 kHz to 1 Hz, with a pulse amplitude of 10 mV. Before the EIS experiments, cyclic voltammograms of 2.5 mmol L^−1^ [Fe(CN)_6_]^4−/3−^ in 0.1 mol L^−1^ KCl and 2.5 mmol L^−1^ [Ru(NH_3_)_6_]^2+/3+^ in 0.1 mol L^−1^ KCl were obtained. The Δ*E*_1/2_ potentials were calculated for each sensor type as half of the difference between the oxidation and reduction peak potentials. The Δ*E*_1/2_ values were then applied as the initial potentials in EIS for each particular sensor type. Five electrodes of each type were used, and three repeated measurements were performed on each one. Values of particular elements in the electrical equivalent circuits (EECs) R([R]/Q) for [Fe(CN)_6_]^4−/3−^ redox marker (Figure 2A) and R([RW]/Q) for [Ru(NH_3_)_6_]^2+/3+^ (Figure 2B) were calculated using FRA simulation in Metrohm NOVA 2.1.5.

#### 2.3.3. Voltammetric Analysis of Lornoxicam

CV was used to compare the voltammetric behavior of lornoxicam on the tested sensors. The following parameters were used: *E*_in_ = −1000 mV, *E*_switch_ = +2200 mV, *v* = 100 mV s^−1^. DPV was applied for the analysis of model solutions and determination of lornoxicam: electrolyte—BRB (pH 3), pulse amplitude = +60 mV, pulse width = 30 ms, *v* = 40 mV s^−1^.

## 3. Results and Discussion

### 3.1. Surface/Material Characterization

The SEM image (Figure 1) shows the BDD thin film grown on the Al_2_O_3_ substrate. The observed grain size is in the sub-microcrystalline range from about 0.2 to 1 μm. The thickness after 30 h of growth is about 3.5 μm and provides a homogeneous covering of the ceramic substrate without pinholes and low electrical resistance.

Raman spectra acquired using 325 nm and 633 nm excitation wavelengths (Figure 2) contain maxima typical for heavily boron-doped diamond. Both B_1_ (480 cm^−1^) and B_2_ (1220 cm^−1^) relate to the boron incorporation and were connected with the acoustical and optical phonon confinement due to changes in the diamond structure, respectively [42]. The ZCP_D_ maximum belongs to the zone center phonon mode of cubic diamond [43] and is shifted to about 1305 cm^−1^ due to the “Fano” effect, i.e., the quantum interference between the zone center optical phonon and a continuum of electronic transitions around the same energy [44], from its original position for a single crystalline diamond at 1333 cm^−1^. While the visibility of Raman bands in the 633 nm spectrum is significantly affected by the “Fano” effect, the 325 nm spectrum further shows a small aC maximum at 1360 cm^−1^ corresponding to the amorphous carbon in the layer, and the G band (graphite band) at 1570 cm^−1^ associated with sp^2^ bonded carbon [45].

The boron doping level and electrical properties were determined by Hall measurement, which reveals the amount of electrically active boron atoms that actively participate in the electrochemical reactions. The recorded concentration was 2.9 × 10^21^ cm^−3^, which is much higher than the semi- to metallic conductivity transition threshold value at 3 × 10^20^ cm^−3^. The corresponding electrical resistivity was 1.7 × 10^−2^ Ω.cm. Following the work of Bernard et al. [46], the boron doping level calculated using the position of B_1_ maximum was 1.7 × 10^21^ cm^−3^.

### 3.2. Electrochemical Characterization

#### 3.2.1. Cyclic Voltammetry

The electrochemical properties of tested LM-SP/BDDEs were investigated by applying cyclic voltammetry (CV), and the obtained results were compared with those achieved using standard electrochemical cell arrangement, including bulk BDDE and commercially available SP/BDDE. The first tested parameter was the width of the usable potential window of individual sensors in the supporting electrolyte of 0.1 mol L^−1^ H_2_SO_4_ (*v* = 100 mV s^−1^). The anodic and cathodic potential limit was defined as the potential, where the current density passed the value ±2000 nA mm^−2^. Table 1 shows that all systems provide a comparable very wide potential window (>3000 mV). However, the widest potential range was observed for the bulk BDDE, corresponding to its lower boron-doping level. Table 1 also shows a shift of hundreds of millivolts in the cathodic direction of SP/BDDE potential window, compared to other tested BDDEs.

Two common redox systems, [Fe(CN)_6_]^4−/3−^ and [Ru(NH_3_)_6_]^2+^/^3+^, were employed for subsequent experiments for characterization of the working electrodes. While [Fe(CN)_6_]^4−/3−^ represents the redox system based on BDDE inner sphere reaction (in the case of sp^2^ carbon and metal electrodes it belongs to the outer sphere redox systems) [13,17,47,48,49,50], [Ru(NH_3_)_6_]^2+^/^3+^ belongs to the outer sphere redox markers [17,48,50,51]. For outer-sphere redox markers, the electron is transferred rapidly through the solvent monolayer; and the reactants, intermediates, and reaction products do not show strong interactions with the electrode surface. Electron transfer is not affected by electrode material or its quality. On the other hand, in inner sphere redox systems, either the reactants, intermediates, reaction products, or in combination, strongly interact with the electrode surface, to which they are often directly adsorbed. Electron transfer is strongly influenced by the condition of the electrode surface—the electrode material. Reactions with the “inner sphere” mechanism are generally less reversible than “outer sphere” reactions [52].

For most of the presented results, the current values were recalculated to current densities with respect to the different surface areas of the diamond working electrodes (a geometric area was used for this recalculation). First, the repeatability of the measurements was tested by measuring 10 cyclic voltammograms of both redox markers in the solution of 0.1 mol L^−1^ KCl applying all investigated sensors (*v* = 100 mV s^−1^, *c*([Fe(CN)_6_]^4−/3−^) = *c*(Ru(NH_3_)_6_]^2+/3+^) = 6.25 × 10^−4^ mol L^−1^). The obtained curves are depicted in Appendix A (Fe) and Appendix A (Ru) and testify to very good repeatability in all cases (relative standard deviation, RSD_10_ < 2.5%).

The reversibility of the used redox markers was examined next. Figure 3A shows cyclic voltammograms of [Fe(CN)_6_]^4−/3−^ recorded at particular electrodes. This is a comparison of the tenth curves from the previously mentioned experiments. Particular values of the anodic and cathodic peak heights (*j*_pa_ and *j*_pc_), their ratio (*j*_pa_/*j*_pc_), and values of the anodic and cathodic peak potential (*E*_pa_ and *E*_pc_) and their difference (Δ*E*_p_) are summarized in Table 2. The achieved values of current densities (*j*_p_) were similar, yet the highest oxidation and reduction peaks were recorded for 2LM- and 3LM-SP/BDDE. The parameter *j*_pa_/*j*_pc_ ranged from 1.03 (BDDE) to 1.18 (SP/BDDE). The values limited to the theoretical value 1 confirming reversibility of the electrode reaction were found, in addition to BDDE (1.03), also for 2LM- (1.07) and 3LM-SP/BDDE (1.06). On the contrary, the potential difference, which is the most important criteria for assessment of the reversibility, was relatively high and was not limited to the theoretical value of 59 mV. This is probably due to the inner sphere nature of the ongoing redox reaction, which is significantly affected by the surface quality of the electrode. The worst reversibility (Δ*E*_p_ = 395 mV) was obtained at the commercially printed sensor, while the Δ*E*_p_ was much lower for both new laboratory-prepared printed sensors and BDDE. The best reversible signals were recorded for 3LM-SP/BDDE (Δ*E*_p_ = 115 mV).

Last but not least, it is necessary to mention the significant potential shift of redox signals obtained on printed sensors, both commercial and laboratory-prepared, to less positive values in comparison with the classical three-electrode arrangement of an electrochemical cell. As documented in Appendix A, this shift is caused by the pseudo-reference electrodes used in the case of SPEs. The figure shows the example of 3LM-SP/BDDE; after connecting an external reference electrode instead of the printed one, the curve shifted, overlapping with the curve obtained on BDDE with the same reference electrode. A similar potential shift was observed throughout the work in the analysis of redox markers as well as other analytes and will not be given further attention in the text.

Similar experiments for the Ru complex are documented in Figure 3B, and the basic parameters of the obtained curves are summarized in Table 3. The parameter *j*_pa_/*j*_pc_ ranged from 0.96 to 1.03 at all tested sensors, limiting to the theoretical value 1 and confirming reversibility of the electrode reaction. In contrast to [Fe(CN)_6_]^4−/3−^, in the case of [Ru(NH_3_)_6_]^2+/3+^, Δ*E*_p_ values were found close to the theoretical value of 59 mV which is typical for the one-electron reversible electrode reaction. The lower values of Δ*E*_p_ for this redox probe could be explained by the nature of the electrode reaction. The outer sphere reaction pathway and the electron transfer as well is in general not influenced by the physicochemical properties of the working surface of the electrode [53]. Excellent reversibility was confirmed, especially for LM-SP/BDDEs and classical BDDE (Δ*E*_p_ = 59–65 mV). For a commercial sensor, the potential difference was again greater (Δ*E*_p_ = 85 mV) compared to others.

The influence of the scan rate on voltammetric responses of both redox markers was investigated in the next step. This parameter varied from 25 to 500 mV s^−1^, and the obtained cyclic voltammograms are depicted in Appendix A for [Fe(CN)_6_]^4−/3−^ and Appendix A for [Ru(NH_3_)_6_]^2+/3+^. It is evident that the signals of both complexes increase with increasing scan rate, but the corresponding dependencies were non-linear. On the contrary, linearity was observed for the dependences between *j*_p_ and the square root of the scan rate (*v*^1/2^), which is illustrated in Figure 4. This result is typical for the diffusion-controlled electrode reaction, which is common for BDDEs due to their low ability to adsorb analytes on the working surface. The statistical parameters of the particular equations (slopes, intercepts, and correlation coefficients) are summarized in Table 4. The highest slope values were again obtained for 2LM- and 3LM-SP/BDDEs, which suggests the excellent electrochemical properties of these sensors. The diffusion-controlled pathway of the observed electrode reactions was further confirmed by the log(*j*_p_)_log(*v*) analysis, particularly by the slopes (Appendix A) of these linear dependencies. They are close to the theoretical value of 0.5, especially in the case of Ru complex (0.4691–0.5255), an outer sphere redox marker, for which the diffusion-controlled reaction is typical.

The apparent heterogeneous electron-transfer rate constants, *k*^0^_app_, were calculated according to Nicholson [54] for the scan rate of 100 mV s^−1^. Based on the literature, the following values of diffusion coefficients (*D*) were used assuming *D*_ox_ = *D*_red_: 7.6 × 10^−6^ cm^2^ s^−1^ [55,56] for [Fe(CN)_6_]^4−/3−^ and 5.5 × 10^−6^ cm^2^ s^−1^ [57] for [Ru(NH_3_)_6_]^2+/3+^. The rate constants are referred to as apparent because no correction for any electric double layer effects was made. The obtained results are summarized in Table 4 and can serve for qualitative estimation of charge transfer. Due to the high values of Δ*E*_p_ in the case of [Fe(CN)_6_]^4−/3−^, the previously published supplemented calculation procedure according to paper [18] was applied. The *k*^0^_app_ values span over three orders of magnitude (1.12 × 10^−3^–3.80 × 10^−5^ cm s^−1^), confirming the surface-sensitive character of this marker. Comparable high values of the rate constant, indicating faster heterogeneous electron transfer, were calculated for LM-SP/BDDEs (highest for 3LM-SP/BDDE) and BDDE. A significantly lower value was obtained for SP/BDDE. The values of *k*^0^_app_ calculated for [Ru(NH_3_)_6_]^2+/3+^ range in the narrow region from 2.98 × 10^−3^ to 2.14 × 10^−3^ cm s^−1^, which corresponds to the inner sphere character of this redox marker and is consistent with previously published results [17,18,55]. The highest values confirming favorable electrochemical properties of the working electrodes were obtained again for 2LM- and 3LM-SP/BDDEs.

#### 3.2.2. Electrochemical Impedance Spectroscopy

All obtained EIS spectra (Figure 5) correlate well with cyclic voltammetry, as the sharpest peaks with the best reversibility were obtained at electrodes with the lowest *R*_CT_ values. The values of the particular elements of the electrical equivalent circuits (EECs) (R([R]/Q) for the Fe redox marker and R([RW]/Q) for the Ru redox marker calculated using the FRA simulation software are summarized in Table 5.

For the [Fe(CN)_6_]^4−/3−^ redox marker (Figure 5A), the highest *R*_CT_ = 40.1 kΩ was recorded on SP/BDDE, corresponding to the largest recorded Δ*E*_p_ between the cyclovoltammetric oxidation and reduction peak potentials. In the case of LM-SP sensors, *R*_CT_ decreased with increasing area of the working electrode, being 21.6, 7.7, and 2.4 kΩ for 1.0, 2.0, and 3.0 mm electrode diameter, or 0.17, 0.24, and 0.17 kΩ cm^2^ when expressed with regards to the electrode area. *R*_CT_ at classical BDDE and commercial SP/BDDE were 11.7 kΩ/0.83 kΩ cm^2^, and 40.1 kΩ/4.1 kΩ cm^2^, respectively.

The *R*_CT_ of [Ru(NH_3_)_6_]^2+/3+^ redox complex (Figure 5B) at classical BDDE and commercial SP/BDDE were much lower, amounting to 1.0 kΩ/71.8 Ω cm^2^, and 0.12 kΩ/12.3 Ω cm^2,^ respectively. At LM-SP/BDDEs, *R*_CT_s were 1.19, 0.94, and 1.98 Ω cm^2^ at electrodes with 1.0, 2.0, and 3.0 mm diameter. The slightly wider range of values was caused by very low obtained *R*_CTs_ at levels of 30 Ω for 2LM/SPBDDE and 3LM/SPBDDE due to difficult EEC fit, as only a few viable points in Nyquist plot semicircles. Lower *R*_CT_s for both used redox probes were observed at electrodes with higher boron doping, which was expected regarding the analogous and well-known cyclovoltammetric behavior of these redox probes at BDDEs with varied boron-doping levels. Furthermore, the consistently low values of obtained *R*_CTs_ at the tested electrodes illustrate the long-term stability of H-termination, as the corresponding *R*_CT_s at O-terminated BDDEs are increased dramatically, as reported by Oliviera and Oliveira-Brett [58].

### 3.3. Application

The application possibilities of the newly printed sensors with working BDDE were tested in the analysis of several bioactive substances. Lornoxicam (LRX, 6-Chloro-4-hydroxy-2-methyl-N-2-pyridinyl-2H-thieno [2,3-e]-1,2-thiazine-3-carboxamide 1,1-dioxide, CAS: 70374-39-9) will be exemplified in this work. The structural formula of this substance is shown in Figure 3. It is a non-steroidal anti-inflammatory drug from the oxicam class (NSAID). The mechanism of LRX effect is predominantly related to the inhibition of prostaglandin synthesis leading to the suppression of inflammation. It causes short-term relief of acute and moderate pain and symptomatic relief of pain and inflammation in osteoarthritis and rheumatoid arthritis. It is also used in postoperative pain management [59,60,61].

#### 3.3.1. Voltammetric Behavior of Lornoxicam and Mechanism of Its Electrochemical Oxidation

Figure 6A shows a comparison of the cyclic voltammograms of 10 µmol L^−1^ LRX recorded on BDDE, SP/BDDE, and 2LM-SP/BDDE in the solution of BRB (pH 3) after conversion to current densities. In accordance with previously published works, one anodic peak was observed, corresponding to the oxidation of the hydroxyl group on the thiazine ring [62,63,64]. The current density values for the oxidation peak of LRX are comparable at all tested electrodes. Only the potential shift described above caused by the use of pseudo-reference electrodes in printed sensors was observed (Appendix A).

The mechanism of the electrochemical oxidation of lornoxicam was studied using online electrochemistry with mass spectrometry (EC-MS) and bulk electrolysis of LRX (*c* = 0.5 mmol L^−1^) on the carbon fiber brush electrode in an aqueous solution of 0.2 mol L^−1^ CH_3_COOH with acetonitrile (1:1, *v*/*v*) at a potential of 800 mV vs. SCE for 1 h, followed by LC-MS analysis of reaction products. Based on the obtained results (Appendix A), a reaction mechanism (Figure 4) was proposed. The electrochemical reaction of LRX (**I**) starts with two-electron oxidation and deprotonation of the hydroxyl group to form unstable intermediate **II**. Nucleophilic addition of water to the cation **II** and ring-chain tautomerism leads to structures **III**, **IV**, and **V**. Under acidic conditions, condensation of **V** gives **VI.** Intermediate V irreversibly decomposes under the C-C bond cleavage to form **VII** and **VIII**. Stable products, the carboxylic acids **IX** and **X** arise from the following electrochemical oxidation of **VII** and acid-catalyzed hydrolysis of **VIII**, respectively. The reaction pathway is analogous to these of the structurally similar compound meloxicam [63].

#### 3.3.2. Determination of Lornoxicam in Model Solutions

Subsequently, parameters of differential pulse voltammetry (DPV) were optimized using BDDE for LRX determination in model solutions. The following conditions were chosen as optimal: supporting electrolyte—BRB (pH 3), *E*_in_ = 0 mV, *E*_fin_ = +2000 mV, pulse amplitude = +60 mV, pulse width = 30 ms (+ additional 20 ms as the current sampling time), *v* = 40 mV s^−1^. The performed optimization experiments are shown in Appendix A. The proposed DPV method was then applied for analysis of model solutions containing LRX using all tested sensors. First, the concentration dependence on the individual sensors was measured. Figure 6B shows an example of DP voltammograms recorded on 2LM-SP/BDDE in the concentration range of LRX from 0.1 to 35 µmol L^−1^, which represents the linear dynamic range (LDR) for this electrode where the peak height increases linearly with the analyte concentration (Figure 6C). The LDRs obtained for BDDE, SP/BDDE, as well as LM-SP/BDDEs are compared in Figure 6D, and the relevant statistical parameters are summarized in Table 6. It can be seen in the figure that all electrodes provide a wide LDR, but while in the case of LM-SP/BDDE the concentration dependence was always linear over the whole range, in the case of both BDDE and SP/BDDE this dependence was divided into two distinct linear sections. In the case of SP/BDDE, it was possible to measure lower LRX concentrations, but on the contrary, at higher concentrations, LDR was terminated much earlier. The achieved LOD values were comparable for all sensors, which again indicates very good electrochemical properties of tested lab-made sensors.

The DPV method for LRX determination was verified by analyzing model solutions spiked with an analyte, and a real sample of the pharmaceutical preparation. Again, the results obtained for all tested sensors were compared. First, model solutions were analyzed. It was distilled water with BRB solution (pH 3) to which a standard drug solution was added so that the final LRX concentration was 1 × 10^−6^ mol L^−1^. The calibration dependence method was used to evaluate the analysis, the determination was repeated 5 times for each sensor, and the corresponding statistical parameters, such as average concentration with the appropriate confidence interval, recovery, and relative standard deviation, were calculated. The results are summarized in Table 7. Subsequently, the pharmaceutical preparation XefoRapid 8 mg (Takeda Austria GmbH, Wien, Austria) was analyzed. Ten tablets were ground in a mortar, the powder was mixed, weighed, and one-tenth of the weight was dissolved in 50 mL of acetonitrile using ultrasound. From the solution thus prepared, 50 µL was pipetted into the electrochemical cell to 10 mL of BRB solution. Again, the calibration dependence method was used, and the obtained results of the analyses are shown in Table 7. The table shows that all tested sensors allow correct and well-repeatable results.

#### 3.3.3. Intra- and Inter-Electrode Repeatability

One of the very important properties of printed sensors is their intra- and inter-electrode repeatability. Intra-electrode repeatability is a parameter that indicates the repeatability of measurements on one electrode and is important for determining the stability of the sensor, its life, and the possibility of its repeated use. Inter-electrode repeatability indicates the repeatability of measurements between individual sensors of the same type and is absolutely essential for so-called disposable sensors, where each particular sensor should provide a reproducible response under the same conditions.

Figure 7A–C shows DP voltammograms of LRX with a concentration of 10 µmol L^−1^ recorded under the optimized conditions, where each curve corresponds to one piece of a sensor of a given type. Because more sensors were available for commercial SPEs, 10 pieces were included in the study, compared to only 5 for LM-SPEs and BDDEs. Each measurement was repeated 10 times, and the figure always shows the 10th curve. The evaluation of peak heights in current density values obtained at individual electrodes is documented in Figure 7D. It is clear that the best inter-electrode reproducibility was confirmed for LM-SP/BDDEs (RSD_5_ ≤ 5.1%). Slightly worse results were obtained for BDDE in the classical arrangement (RSD_5_ = 11.8%), where, however, use as a disposable sensor cannot be expected, and the intra-electrode repeatability is the more important parameter. The worst repeatability between electrodes was obtained for SP/BDDE (RSD_10_ = 18.8%). Simultaneously, the instability of the potential of LRX oxidation maximum is evident in Figure 7B when using SP/BDDEs, which is most probably caused by the instability of the pseudo-reference Ag electrode. In contrast, the potential of the LRX peak was essentially unchanged both in the case of the classical arrangement with BDDEs in combination with Ag|AgCl(KCl sat.) and in the case of the tested LM-SP/BDDEs with Ag|AgCl reference electrode which seems to be sufficiently stable.

The intra-electrode repeatability is illustrated by the error bars for the individual columns representing the particular sensors in Figure 7D. Again, it is clear that very good results were achieved for LM-SP/BDDEs, especially for 1LM-SP/BDDE (RSD_5_ ≤ 0.65%) and 2LM-SP/BDDE (RSD_5_ ≤ 0.44%), which indicates the very good stability of these new sensors and also predetermines them for re-use as with common bulk BDDEs. As expected, equally good results were achieved for BDDEs (RSD_5_ ≤ 0.59%), which is intended for long-term re-use, and good intra-electrode repeatability is, therefore, a must. Intra-electrode repeatability was also more than acceptable when repeatedly measuring LRX with SP/BDDE.

## 4. Conclusions

In this work, novel screen-printed sensors with chemically deposited boron-doped diamond electrodes were prepared and characterized using scanning electron microscopy and Raman spectroscopy. The electrochemical properties were investigated using cyclic voltammetry of inner-sphere ([Fe(CN)_6_]^4−/3−^) and outer-sphere ([Ru(NH_3_)_6_]^2+/3+^) redox markers. It was confirmed that the results obtained for newly printed sensors were comparable and, in some parameters, even better than for bulk BDDE in the classical arrangement of the electrochemical cell. Compared to commercially printed sensors, in most cases, our laboratory-made devices showed better results. The application possibilities of all of the tested sensors were verified in the analysis of the anti-inflammatory drug lornoxicam. LM-SP/BDDEs again showed very good statistical parameters, such as wide LDRs and low LODs at the levels of 10^−7^ mol L^−1^. Moreover, the presented results show that the novel LM-SP/BDDEs provide very good intra- as well as inter-electrode repeatability and can serve as disposable as well as reusable sensors. Based on our experience, it was possible to work with one sensor for several weeks without visible changes in the height or shape of the observed current signals. Lastly, the electrooxidation mechanism of lornoxicam was elucidated to provide comprehensive background for practical application of these sensors.

## Data Availability

Data are available on request from the corresponding authors.

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
