# Peer review of "Novel Screen-Printed Sensor with Chemically Deposited Boron-Doped Diamond Electrode: Preparation, Characterization, and Application"

_biosensors, 2022, doi:10.3390/bios12040241_

Round 1

Reviewer 1 Report

In this paper, the authors developed novel screen-printed sensors with boron-doped diamond electrodes, produced by a combination of two techniques, a large-area linear antenna microwave chemical vapor deposition system and screen-printing technique. The new sensors exhibited better performance than bulk BDDE in the classical arrangement of electrochemical cell and commercial printed sensors. In addition, it was successfully applied for the detection of lornoxicam. Overall, the results in this manuscript are meaningful, but there are many issues that should be resolved. Therefore, I suggest the acceptance of the manuscript after accommodating the following comments.  

  1. In the material characterization part, the authors need to add one or two references.
  2. The authors need to explain some terms such as fano effect, Z band, ZCPD, etc. In addition, the authors need to check the whole manuscript, correct the mistakes, and get their manuscript edited by the native speakers.
  3. The new electrode (LM-SP/BDDE) and other commercial screen-printed electrodes should be compared and summarized in table to emphasize the significance of this work.
  4. The authors need to estimate the surface area of working electrode (LM-SP/BDDE). In addition, the authors need to calculate the limit of detection, which should be compared to the ones obtained in the commercial SPE (SP/BDDE).
  5. The authors need to demonstrate the detection of lornoxicam in real samples. If it is difficult to get the real samples, the authors need to perform the spike-and-recovery experiments.
  6. The authors only used lornoxicam to verify the application. How about other substances? In the abstract, it was written that several bioactive substances important in the field of environmental protection or human health were tested by these new sensors. Thus, the authors need to demonstrate that this newly prepared electrode can be applied to the sensitive detection of DNA, or other small biomolecules.

Reviewer 2 Report

The present manuscript reported Novel screen-printed sensor with chemically deposited boron-2 doped diamond electrode: preparation, characterization, and application. All aspects are investigated and enough experiments have been done. I believe it is suitable for publication in Biosensors journal after minor revision. Here my comments

  1. in the abstract, what does it mean? “high hardness make”, it is not clear.
  2. in the abstract: “resistance to passivation”, but you did not check it, please add more information or remove it.
  3. please add more information, about “inner sphere “ and “ outer sphere “, conventional inner sphere ([Fe(CN)6]4−/3−) and outer sphere ([Ru(NH3)6]2+/3+) redox. What does it mean?
  4. page 4, line 164, please check the ratio: 312 500 ppm ! is it correct?
  5. please add more information about the mechanism of electrochemical oxidation of lornoxicam at the electrode surface.
  6. page 8, please add more explanations about shifting potential in Cyclic voltammograms in Figure 3.
  7. Table 5, why the Rs amount for “SP/BDDE” is high? Or why is changing? (Rs: solution resistance).
  8. Some papers about the application of Screen-printed electrodes, please add them to your paper.

-Costa-Rama, E., & Fernández-Abedul, M. T. (2021). based screen-printed electrodes: A new generation of low-cost electroanalytical platforms. Biosensors, 11(2), 51.

- Hatamie, A., Rahmati, R., Rezvani, E., Angizi, S., & Simchi, A. (2019). Yttrium hexacyanoferrate microflowers on freestanding three-dimensional graphene substrates for ascorbic acid detection. ACS Applied Nano Materials, 2(4), 2212-2221.

-Saenchoopa, A., Klangphukhiew, S., Somsub, R., Talodthaisong, C., Patramanon, R., Daduang, J., Daduang, S. and Kulchat, S., 2021. A Disposable Electrochemical Biosensor Based on Screen-Printed Carbon Electrodes Modified with Silver Nanowires/HPMC/Chitosan/Urease for the Detection of Mercury (II) in Water. Biosensors, 11(10), p.351.

-Tyszczuk-Rotko, Katarzyna, Jędrzej Kozak, and Bożena Czech. "Screen-Printed Voltammetric Sensors—Tools for Environmental Water Monitoring of Painkillers." Sensors 22, no. 7 (2022): 2437.

- Antuña-Jiménez D, González-García MB, Hernández-Santos D, Fanjul-Bolado P. Screen-printed electrodes modified with metal nanoparticles for small molecule sensing. Biosensors. 2020 Feb;10(2):9.

Round 2

Reviewer 1 Report

The authors addressed all the issues raised by the reviewers and I am satisfied with the updates. Therefore, I suggest the acceptance of this manuscript.

Reviewer 2 Report

Dear Editor

The new version has been improved . please ACCEPT!!